Event-driven industrial robot control architecture for the Adept V+ platform

Semeniuta Oleksandr oleksandr.semeniuta@ntnu.no 1
Falkman Petter 2
1 Department of Manufacturing and Civil Engineering, NTNU Norwegian University of Science and Technology , Gjøvik , Norway
2 Department of Electrical Engineering, Chalmers University of Technology , Gothenburg , Sweden
Dolev Shlomi
Electronic publication date: 2019 Jul 29
Publication date: 2019
Volume: 5
Electronic Location ID: e207
Received 2019 Feb 22; Accepted 2019 Jun 29
Copyright: ©2019 Semeniuta and Falkman
Copyright year: 2019
Copyright holder: Semeniuta and Falkman
License: This is an open access article distributed under the terms of the Creative Commons Attribution License, which permits unrestricted use, distribution, reproduction and adaptation in any medium and for any purpose provided that it is properly attributed. For attribution, the original author(s), title, publication source (PeerJ Computer Science) and either DOI or URL of the article must be cited.
License URL: https://creativecommons.org/licenses/by/4.0/

Keywords: Robotics, Adept, Coroutines, AsyncIO, ZeroMQ, Robot architecture, Computer vision, Communication protocols, Concurrency, System composability

Funding: Norwegian Research Council through the MultiMat project and SFI Manufacturing This paper was financed by the Norwegian Research Council through the MultiMat project and SFI Manufacturing. The funders had no role in study design, data collection and analysis, decision to publish, or preparation of the manuscript.

==============================
Modern industrial robotic systems are highly interconnected. They operate in a distributed environment and communicate with sensors, computer vision systems, mechatronic devices, and computational components. On the fundamental level, communication and coordination between all parties in such distributed system are characterized by discrete event behavior. The latter is largely attributed to the specifics of communication over the network, which, in terms, facilitates asynchronous programming and explicit event handling. In addition, on the conceptual level, events are an important building block for realizing reactivity and coordination. Event-driven architecture has manifested its effectiveness for building loosely-coupled systems based on publish-subscribe middleware, either general-purpose or robotic-oriented. Despite all the advances in middleware, industrial robots remain difficult to program in context of distributed systems, to a large extent due to the limitation of the native robot platforms. This paper proposes an architecture for flexible event-based control of industrial robots based on the Adept V+ platform. The architecture is based on the robot controller providing a TCP/IP server and a collection of robot skills, and a high-level control module deployed to a dedicated computing device. The control module possesses bidirectional communication with the robot controller and publish/subscribe messaging with external systems. It is programmed in asynchronous style using pyadept, a Python library based on Python coroutines, AsyncIO event loop and ZeroMQ middleware. The proposed solution facilitates integration of Adept robots into distributed environments and building more flexible robotic solutions with event-based logic.

Introduction

Robots are always associated with high level of complexity, which is usually considered with respect to the task being performed. Because modern robotic systems operate in a networked environment, the additional complexity is caused by the nature of interaction and coordination with sensors, vision systems, various mechatronic equipment, and computational components, such as servers and cloud services. As Kortenkamp & Simmons (2008) note, “robot systems need to interact asynchronously, in real time, with an uncertain, often dynamic, environment. In addition, many robot systems need to respond at varying temporal scopes—from millisecond feedback control to minutes, or hours, for complex tasks”. This is the reason why robotic architecture becomes increasingly important.

To tackle the challenges associated with communication and coordination complexity, a number of robotic middleware solutions has been developed, most notably, the Robot Operating System (ROS). Most of them support publish—subscribe messaging style, where messages are asynchronously delivered from publisher to subscriber nodes.

The primary use of the publish/subscribe mechanisms in systems based on robotics middleware is delivery of periodic sensor readings. The reason is that in many cases, especially in the research environment, robots are viewed as continuous systems, characterized by a set of continuous time-varying signals, sampled at a constant interval (Dantam et al., 2016). The idea of periodic sampling is central to the classical approach to digital system control, as it is rooted in the solid theory of periodic control (Astrom, 2008). Since robotics shares its intellectual tradition with control engineering, periodic sampling became a dominant strategy for real-time sensing in robotic systems. Although the periodic approach works well for traditional point-to-point control systems, in the distributed environment one faces such challenges as latency, jitter in packet delivery, and lost packets. In addition, with introduction of more resource-constrained devices, the cost of communication increases, and it becomes of critical interest to base the control system on reactivity only to events of particular importance (Miskowicz, 2015). In addition to events derived from continuous signals (e.g., based on a signal threshold), numerous classes of sporadic discrete events constitute an important abstraction for modeling behaviors in cyber-physical systems.

The discrete event behavior is apparent in situations when a system is convenient to model as possessing discrete state space (e.g., {IDLE, BUSY, DOWN}), when a system automates discrete parts manufacturing, when human iteration is involved (e.g., button press at an arbitrary time), when an unpredictable disturbance requires a system’s component restart, and many others. For a number of automated components (e.g., robots, feeders etc.) cooperating, events constitute an important abstraction that ensures synchronization of the components’ operation. In addition to the inherent discrete event nature of many processes where industrial robots are involved, latency, attributed to any communication system, calls for event-driven and asynchronous design of computational modules that involve networking.

From the engineering perspective, it is beneficial to be able to compose a robot control program from modular well-defined components, rather than coding up everything from scratch in a monolithic fashion (Onori & Oliveira, 2010). One approach is to specify sequences of operations to form serial, parallel, or arbitrary order sequences (Lennartson et al., 2010). This paper makes use of the coroutine abstraction to define composable communication-heavy tasks with event-based logic. The main idea is to treat the robot and external nodes (such as vision systems) as services, and coordinate communication with them from a high-level control node. The logic of the latter is composed from a set of Python AsyncIO coroutines based on domain-specific abstractions. The benefit of this approach is that one is able to realize much more complex coordination scenarios, where the burden of error-prone communication is lifted from the native yet restricted robot controller platform to a higher-level AsyncIO-driven application. This paper applies the described line of thinking as a custom solution for the Adept V+ robot platform. Although the described solution is platform-specific, the paper aims at establishing a common design paradigm for multitude of industrial robots.

At the core of the proposed robot control architecture lies two-sided communication: (1) TCP/IP connection between the robot controller and the control node, and (2) publish–subscribe communication with the external nodes. The architecture facilitates specification of event-driven communication logic and internal asynchronous runtime. As a result, it becomes easier, more flexible, and less error-prone to program industrial robots and integrate them into distributed environments. The implementation of the system is done based on AsyncIO Python abstractions and ZeroMQ middleware. The architecture is validated by a proof-of-concept implementation of a robotic system with vision sensing, where a robot, a set of GigE Vision cameras, and a set of computing nodes are connected in a VLAN-based network configuration.

The paper is structured as follows. First, the background on approaches to robot programming, implementation and semantics of events in computational systems, and event-driven middleware is presented. Further, the principles and architecture of the proposed system are described. An application use case that validates the ideas of this paper is then introduced and analyzed. The proposed solutions are discussed and compared to similar systems. The paper concludes with outlining the further development directions.

Background

Approaches to programming industrial robots

Industrial robots are supplied as end-products with carefully engineered hardware and software components. As robots are essentially programmable devices, an important part of the system is the robot controller with the associated computational capabilities and programming interface. An Adept robot controller runs V+ (Adept Technology, Inc., 2014), a real-time multi-tasking operating system. It controls robot motion, input/output, task management, and other necessary operations. V+ is also the name of a programming language for the Adept platform. Other robot platforms are based on similar operating systems, e.g., KUKA.SystemSoftware and ABB RobotWare.

When developing a robot control program, the most straightforward way to go is programming all the logic to be run by the robot controller. The availability of various I/O interfaces, such as digital signals, DeviceNet, Ethernet, and RS-232, allows to communicate with external systems when needed. In such an arrangement, the native robot controller has the master role. To program an Adept robot controller, one requires a client programming environment (such as Adept ACE, Adept DeskTop), that runs on a Windows-based programming station. V+ codebase is comprised of subroutines called programs that are gathered in modules. An individual program can be associated with an operating system task, which in turn can be run concurrently with other tasks. Figure 1 shows this kind of configuration, with task 1 being associated with the V+ program highlighted yellow.

Figure 1 Classical robot programming setup in context of the Adept platform.

The configuration where the robot has the master role offers good timing properties, but makes the control logic rather rigid and monolithic. Specifically, it becomes difficult to operate such robot as a part of a system with a large number of distributed components. When more flexibility is required, an alternative solution is a client/server configuration, where the robot controller runs a TCP/IP server, and accepts commands from a client on a general-purpose platform (e.g., x86-64- or ARM-based computer running Linux). The client can be more flexibly programmed, and normally constitutes a part of a larger distributed architecture, for example, as a ROS node or other computational component. The defined set of commands, may include various motion tasks, as well as coarse-grained tasks, pre-defined as V+ programs in the robot controller. One of the challenges in this latter case is to define a suitable wire protocol and ensure that commands to the robot controller are read and processed correctly, given the stream nature of the TCP protocol.

Events in computational systems

One distinguishes between time-driven and event-driven systems. In the former case, everything is modeled with respect to a clock with a given frequency. For example, a continuous signal is sampled at a constant time interval. Such systems are a subject of study in classical control. Conversely, event-driven paradigm presumes that a system is characterized by a discrete state space, and events can occur at any time instant. An event in this case constitutes any instantaneous occurrence that causes transition from one system state to another (Cassandras & Lafortune, 2008).

The behavior of industrial automation systems is to a large extent event-driven. It manifests itself in both the nature of the applications and in the inherent properties of digital communication systems, specifically the latency. To deal with the latter, the operating system provides the abstraction of I/O events. An I/O event is associated with a particular resource becoming ready, e.g., data has arrived from the network and is ready for non-blocking access. On *nix platforms with POSIX system call API, mechanisms for monitoring I/O events comprise I/O multiplexing (select/poll), signal-driven I/O, the Linux-specific epoll, and BSD-specific kqueue (Kerrisk, 2010). Windows features the IOCP threading model for concurrent handling of asynchronous I/O.

To tackle system design when waiting for a network operation to complete, the event-driven programming paradigm is employed, which directly harnesses I/O events. As shown in algorithm 1 (adapted from the event-driven server example by Rhodes & Goerzen (2010)), an event driven system component is comprised of (1) a continuous event loop, (2) a polling source p (such as one of the I/O monitoring system calls with a set of monitored file descriptors), (3) a set of possible events on every loop iteration (events), and (4) data containers for accumulation of request and response data (Din, Dout).

___________________________________________________________________________________________________________ Algorithm 1 Event-driven component ___________________________________________________________________________________________________________   1:  function EventLoop(p)   2:     Initialize S                                 ⊳ Set of monitored connections   3:     Initialize Din                     ⊳ Dictionary of input bytes per connection   4:     Initialize Dout                  ⊳ Dictionary of output bytes per connection   5:     while true do   6:        events ← poll(p)      ⊳ Set of events available for processing at this time   7:        for e ∈ events do   8:            if e = New connection c is available then   9:                 Add c to S  10:            else if e = Connection c got closed then  11:                Remove c from S  12:                Remove Din[c] and Dout[c]  13:            else if e = Data available for reading at connection c then  14:                Read from c to Din[c]  15:                if Din[c] contains a complete request then  16:                    Dout[c] ← Process bytes in Din[c]  17:                end if  18:            else if e = Connection c is available for writing then  19:                Write to c some bytes from Dout[c]  20:                Dout[c] ← Unsent bytes in Dout[c]  21:            end if  22:        end for  23:     end while  24:  end function  ______________________________________________________________________________________________________________

Implementing such loops directly is error-prone. Therefore, a number of event-driven networking frameworks exist for different programming languages. They encapsulate the low-level system calls and provide object-oriented interface based on such design patterns as Reactor and Proactor (Buschmann, Henney & Schmidt, 2007). Such patterns realize the Inversion of Control (IoC) principle, where application-specific callbacks are registered, and later called by the framework on occurrence of the respective events. Examples of such frameworks are Boost Asio and ACE for C++, and AsyncIO and Twisted for Python.

AsyncIO (Python Software Foundation, 2019) is a part of Python standard library, introduced in Python 3.4, which provides a standardized way for implementing event loops with sets of concurrent non-blocking coroutines. A coroutine constitutes an executable object that represent a particular application-specific task. An important feature of a coroutine is that it can pause its execution and yield control to the event loop. As a result, one achieves cooperative multitasking, where a set of coroutines get suspended and resumed as different events occur and conditions get satisfied. Such programming style gives the most evident advantage in realizing scalable event-driven servers. In addition, as further shown in this paper, several communication-heavy tasks can be composed together in a well-defined way when implemented as AsyncIO-based coroutines.

Another use case of explicit utilization of events is asynchronous programming. A functions call can be either synchronous, which blocks until the function completes, or asynchronous, which returns immediately. The results of an asynchronous function call shall be processed as a part of other operation once they become ready. A completion event in this case can be captured and processed in a way similar to algorithm 1. As a polling source, a blocking queue can be used, with one execution thread putting a event object in it, and the other thread (running the polling loop) getting an event object from the queue when one is ready. When using future objects, the polling source constitutes a set of futures, which are monitored in a way similar to polling file descriptors in algorithm 1. Although futures and queues are typically used to realize thread-based concurrency, their counterparts exist also in AsyncIO.

Similarly to synchronous vs asynchronous function calls, when dealing with networking, one can realize two communication styles: request/reply, which is logically synchronous, and publish/subscribe, which decouples operations of sending/publishing a message and its receiving on the subscriber’s side. The publish–subscribe messaging in the backbone of event-driven middleware, which is presented in the following subsection.

Event-based middleware

As the contemporary robotic systems are necessarily distributed, an important component is robotic middleware, which provides a unified set of communication and input/output capabilities.

The transport layer functionality (TCP and UDP protocols) on POSIX-based operation systems can be implemented via the C-based socket API, as well as by utilizing various object-oriented network frameworks (Komu et al., 2012). The “native” connection-oriented communication (using a pair of TCP/IP sockets) is a straightforward and low-overhead means of exchanging data between two networked nodes. It, however, presumes temporal dependency, i.e., requirement for the components to be available at the same time, along with address knowledge and agreement on data representation. In contrast, messaging middleware allows to decouple the communicating components by introducing message queuing, built-in address resolution (e.g., via handling logical addresses such as topic names), and usage of a common data serialization format (Magnoni, 2015). An important feature of a middleware is the provision of the publish/subscribe and other messaging patterns, which allows to design a distributed system in an event-driven fashion.

The defining components of a particular middleware solution are the communication protocol (transport-level TCP and UDP, wire-level AMQP, ZeroMQ/ZMTP, MQTT), the communication styles (request/reply, publish/subscribe), and the data serialization method (typically based on an interface definition language like Protobuf or Apache Thrift). Many middleware solutions are based on a central broker, e.g., ActiveMQ and RabbitMQ. The additional hop through the broker adds a constant value to the communication latency (Dworak et al., 2012), which is not desirable for time-sensitive applications such as robotics. ZeroMQ is an example of broker-less middleware, in which the message queuing logic runs locally within each communicating component (ZeroMQ, 2008).

Several state-of-the art middleware solutions has been evaluated at CERN (Dworak et al., 2011) to find the most suitable candidate for realizing the upgraded CERN Controls Middleware, a system responsible for managing communication with a multitude of sensors and actuators at the organization’s accelerator complex. Similarly to the specifics of the robotics context, for CERN engineers it was important to achieve low latency, as well as low memory and resource usage. In addition, it was preferable to possess a solution without message brokers, central servers, or additional daemons. As a result of the performance study with different communication scenarios (Dworak et al., 2012), ZeroMQ was chosen as the most suitable technology. Specifically good results were shown with respect to system scalability (latency was kept relatively constant regardless the number of clients added) due to its automatic buffering capability.

Robot Operating System (ROS) is the most widely used middleware that is specifically designed for building distributed robotic systems. It supports request/reply remote procedure calls via services, and publish/subscribe communication via topics. Messages in ROS are serialized with the built-in serialization mechanism. A ROS system requires a central master server, responsible for name resolution. On the transport layer, both TCP and UDP are supported via standard sockets. In its current form, ROS is tightly coupled with Ubuntu as the runtime platform.

To preserve the philosophy of ROS and most of working code, yet adapt it to the production environment, the ROS2 initiative has started (Gerkey, 2018), introducing cross-platform support (including for small embedded platforms) and built-in real-time control capabilities. Data Distribution Service (DDS) is used as a communication backbone in ROS2. DDS is a data-oriented middleware standard with several industrial-grade implementations, which provides various transport configurations suitable for real-time control (e.g., deadline and fault-tolerance) (Maruyama, Kato & Azumi, 2016).

ROS is perhaps the most widely used robotic middleware, although not the only one. The development of the iCub humanoid platform has spinned-off YARP, which is based on ACE for communication and Thrift for typed data serialization. YARP supports the publish/subscribe messaging paradigm and different buffering policies, such as FIFO and Oldest Packet Drop (ODP) (Natale et al., 2016).

The low-level operating system capabilities are directly utilized by the ach interprocess communication library (Dantam et al., 2015; Dantam et al., 2016), designed specifically for real-time transmission of periodically sampled sensor signals in a robotic system. The goal of the library is to guarantee processing of the latest sample with a minimum latency. Contrary to traditional robotic middleware solutions, ach is implemented as a Linux kernel module.

Robotic middleware solutions such as ROS, YARP, and ach are rooted in the research environment, and often used with complex prototypes such as humanoid robots, AGVs, and various custom-built robotic systems. When it comes to industrial robots, specifically older models, their support is rather limited, even in the context of the popular ROS platform. Support for the latter is being added by the participants of the ROS-Industrial consortium, and, for some new robot models, ROS support comes built-in out of the box. In general, however, there is a long way to go. The Adept platform, for instance, does not have a maintained ROS driver.

The pilot implementation of the robot control solution proposed in this paper uses ZeroMQ as the middleware for publish–subscribe communication. In addition to being highly lightweight, efficient, and cross-platform, ZeroMQ natively supports the AsyncIO event loop.

System Architecture

This section presents the architecture for industrial robot control that facilitates integration of the robots into distributed systems with publish–subscribe communication and building flexible solutions with event-based logic. First, the general principle of the architecture, along with a step-by-step example of communication between the robot controller and the high-level control node is presented. Further, a more detailed description of the proposed wire protocol is described, followed by an overview of the developed software abstractions based on AsyncIO and ZeroMQ.

Components and general principles

The proposed system for control of an industrial robot is based on two principal networked components: (1) RobotServer, a TCP/IP server realized on the robot controller, and (2) MasterControlNode, a high-level control node running on a dedicated computer, perfoming computations aimed at establishing the desired control logic, and communicating with both the RobotServer and other networked components (Fig. 2).

Figure 2 Structure of the proposed robot control system.

RobotServer is associated with the defined set of actions, dubbed skills. They are implemented as procedures in the robot controller’s programming language, each accepting its own set of parameters. To make the robot perform a specific skill, the MasterControlNode sends the corresponding ASCII byte string over the TCP socket connecting it with the RobotServer. A correctly formatted byte string that correspond to the available skill is referred to as a command message, and it is always delimited with “\r\n”. A more formalized description of the communication protocol between MasterControlNode and RobotServer is presented in the following subsection.

As an example, consider a robot operation of motion towards a specified pose in the world coordinate frame. To perform it, MasterControlNode sends a byte string such as the following:

eae86869:move_to:−80.000,−481.000,112.500,180.000,90.000,180.000∖r∖nee861124:break∖r∖n

The above byte string is comprised of two messages, corresponding to skills move_to and break, each defined as a V+ program. The former accepts six real-valued arguments (in this case, x, y, z, yaw, pitch, roll), while the latter is invoked without arguments. It is a common pattern to combine a motion command with breaking, as this ensures that several subsequent motions are not interpolated, and the robot’s end effector reaches the specified pose.

The first 8 bytes of each message correspond to a unique ID, generated by the MasterControlNode. It is obtained as the first 8 bytes of an UUID generated by the Python’s uuid.uuid4 function. The remaining components, separated by a colon, constitute the name of the skill and the list of parameters.

RobotServer operates as a task in the robot controller that realizes a TCP server. After a complete message is read from the TCP stream, it is mapped to the specific skill, which gets executed, with the start and completion timestamps being recorded. After the completion, an acknowledging message of the following form is sent back to MasterControlNode:

eae86869:done:2492.516,2492.539:480.014,−0.038,709.975,0.000,179.995,0.004∖r∖n

The parts of the acknowledging message, separated by colons, constitute the associated command message ID, execution status, starting and completion timestamps separated by a comma, and current robot pose separated by commas. The timestamps are measured in seconds with millisecond precision, and the pose parameters correspond to translation vector components x, y, z, expressed in millimeters, and rotation angles (yaw, pitch, roll) expressed in degrees.

MasterControlNode monitors the arrival of responses from RobotServer, and, for the current operation, once all of the IDs that characterize this operation have been acknowledged, the operation is marked as completed.

In addition to communication with the RobotServer via TCP, the MasterControlNode participates in a distributed publish/subscribe network. All the communication primitives are implemented on top of Python’s AsyncIO event loop.

Communication protocol

MasterControlNode and RobotServer communicate with a TCP-based protocol described in this subsection.

When MasterControlNode (the client) gets connected to RobotServer (the server), a communication session is established. It stays active until one of the endpoints (normally the client) closes the connection. Both endpoints send each other streams of bytes, where every byte is semantically regarded as the corresponding ASCII-encoded character. Sequence “\r\n” has a role of delimiter string, which separates two consecutive messages.

delimiter  ::= "∖r∖n"

The primary workflow in the proposed architecture is that the client sends commands to the server, the latter execute those commands, and sends the response back. As such, messages sent from the client to the server are referred to as command messages. A command message represent a string that can be mapped to a predefined V+ program (skill). Currently, four classes of command messages are defined, responsible namely for (1) motion, (2) braking, (3) air triggering, and (4) setting the speed of the subsequent motion:

command_message  ::=  motion_msg  | break_msg  | air_msg  | speed_msg

Motion messages map to motion skills, which constitute V+ programs having 6 real-valued arguments (representing either poses or joint vectors). Formally, a motion message, along with its components, is defined as follows:

motion_msg  ::=  motion_command_name  ":"  location  delimiter motion_command_name  ::= "move_to"          | "move_joints"      |                          "move_rel_world"   | "move_rel_joints"  | "move_rel_tool" location  ::=  real "," real "," real "," real "," real "," real real  ::=  digit+ "."  fractional; fractional  ::=  digit  digit  digit digit  ::= [0−9]

Semantics of the motion skills is the following:

• move_to:x,y,z,yaw,pitch,roll moves the robot’s tool center point to the specified pose, expressed in the world coordinate frame. Here x,y,z represent a translation vector expressed in millimeters, and yaw,pitch,roll represent a vector of rotation angles expressed in degrees.

• move_rel_world:x,y,z,yaw,pitch,roll performs movement relative to the current pose expressed in the world coordinate frame: given the current pose wξnow, move to the pose wξnow⊕ξ, where ξ is embodied in the pose parameters x,y,z,roll,pitch,yaw.

• move_rel_tool:x,y,z,yaw,pitch,roll performs movement relative to the current pose tξnow expressed in the tool coordinate frame (tξnow⊕ξ).

• move_joints:j1,j2,j3,j4,j5,j6 move robot to the specific joints configuration.

• move_rel_joints:j1,j2,j3,j4,j5,j6 performs relative joint movement by displacing each joint in the amounts specified by j1,j2,j3,j4,j5,j6.

In addition to motion_msg, other command messages are responsible for the following: break_msg signalizes the robot that two subsequent motions with break in between shall not be interpolated; air_msg is used to control vacuum valves attached to the tool; speed_msg specifies a speed factor for the subsequent motion command.

speed_msg  ::= "set_speed" ":" (digit  | digit  digit  | digit  digit  digit)                                 //  speed  factor  range: 0−100 air_msg  ::= ('enable_air'  | 'disable_air ')  delimiter break_msg  ::= "break" delimiter

One or more commands constitute a command chain.

command_message_chain  ::=  command_message+

By the nature of the TCP protocol, the bytes are communicated between endpoints in a streaming fashion: the correct order of bytes is guaranteed, but a sent message may not be delivered as an atomic entity, and may arrive in pieces. That’s why, on both the MasterControlNode’s and RobotServer’s side, buffering of the incoming bytes is performed.

Software abstractions

The pilot implementation of the proposed architecture is released as the pyadept Python library (Semeniuta, 2018b), together with the associated V+ code as the AdeptServer project (Semeniuta, 2018a), both licensed under the 3-clause BSD license. The intended workflow is based on treating the robot controller as a service. The first stage is to enable high power for the robot and start the AdeptServer’s server V+ program as a task on the robot controller. After this, a Python program based on pyadept can be launched and used for the high-level system coordination.

Robot commands (as specified in the “Communication protocol”), are defined as classes in the pyadept.rcommands module. They construct immutable instances providing the functionality of correct generation of the corresponding byte strings. All robot command classes realize the get_messages method, returning a tuple of byte strings, each finalized with the delimiter sequence ”\r\n”.

The pyadept.rprotocol module consists of classes, functions and coroutines realizing the logic behind two-sided communication of a MasterControlNode, as well as tools for data capture during system operation. The two central classes of this module are RobotClient and ProtobufCommunicator, realizing AsyncIO-based communication with the RobotServer and the external systems respectively:

• RobotClient provides coroutine methods connect (establishing the connection with the server), as well as cmdexec and cmdexec_joined (providing execution of commands). The two latter methods accept one on more instances of robot commands and initiate communication with the RobotServer using AsyncIO’s StreamWriter/StreamReader pair. Several commands supplied to cmdexec are handled one-by-one: each command’s messages are sent to the server, and the corresponding responses are awaited before proceeding to the next command. Conversely, cmdexec_joined combines messages from the supplied commands into a single sequence, and sends all of them in one run.

• ProtobufCommunicator uses AsyncIO-compatible ZeroMQ primitives to announce a Protobuf-based request event and wait for the corresponding Protobuf-based response in the context of a publish/subscribe system.

Application Use Case

Problem context

The functionality of the proposed system is demonstrated on a robotic application of handling a small part for a detailed vision-based quality inspection. Drawing from the previous work on picking and inspection of small automotive components (Semeniuta, Dransfeld & Falkman, 2016), the described setup is aimed at moving the part from the pick pose ξpick to the inspection-start pose ξis in front of a Prosilica GC1020C camera with a 35 mm Fujinon HF35HA-1B lens, with the subsequent sequence of tool rotations while keeping the part in focus of the camera. On each rotation increment, an image from the camera is requested by the MasterControlNode.

The primary operation, described in the previous paragraph, is performed after the initial calibration phase, which includes determination of the inspection-start pose ξis. Since the chosen camera setup is aimed at close-range imaging of small parts, it is rather sensitive to the depth at which the manipulated part is being held. That’s why, in order to determine the focus plane, a calibration tool is moved first to the approach pose ξappr, and eventually aligned with the focus plane by a series of small linear motion increments with vision feedback. A robot in poses ξappr and ξis is shown in Fig. 3.

Figure 3 Robot in the approach pose ξappr (A) and the sought inspection-start pose ξis(B).

System components

To realize the described task and demonstrate composability of the MasterControlNode/ RobotServer pair with components providing vision services, the system shown in Fig. 4 is considered. In addition to the robot-related components, it includes an FxIS-based image acquisition service ImAcqService (Semeniuta & Falkman, 2018), and ImProcNode –a computational component responsible for image processing. The ImProcNode component is realized with EPypes primitives (Semeniuta & Falkman, 2019), i.e., the logic of the image processing routine is specified as a directed acyclic bipartite graph of computational procedures as data tokens.

Figure 4 Event-driven communication in a robotic cell.

The vision-related components (ImAcqService and ImProcNode) are composed via thread-based concurrency, with asynchronous communication being performed via blocking queues. ImAcqService performs continuous image acquisition from one or more GigE Vision cameras while keeping a circular buffer of a number of recent images. On arrival of a vision request event, the buffer is queried to retrieve the most closely associated in time image. The latter undergoes processing in the ImProcNode, with the result being published.

More complex systems bearing the same architecture can be comprised of larger number of nodes. The subset of components shown as a part of the gray-shaded region constitutes the system with publish–subscribe communication. The green double-sided arrow shows the TCP-based communication between MasterControlNode and RobotServer.

Network topology design

The laboratory setup used in this paper constitutes a distributed system (Fig. 5) consisting of a robot controller (ROB, Adept V+), three GigE Vision-based industrial cameras (C1, C2, C3), and three computational nodes: a computer performing image acquisition and running computer vision routines (VIS, Ubuntu), a robot programming station running Adept DeskTop (ADT, Windows), and a Raspberry Pi single-board computer running the master control node (MCN, Debian).

Figure 5 Network configuration.

The above components are joined into two IP networks: (1) the Robot Network, and (2) the Vision Network. Both of them are realized with a single managed network switch (Netgear GS108Ev3) supporting the IEEE 802.1Q protocol for virtual LANs. The MCN component is configured with two virtual network adapters and connected to the tagged port on the switch. MCN thus belongs to both networks.

With regards to the components shown in Fig. 4, they are deployed as follows. ROB hosts the RobotServer, while MCN hosts the MasterControlNode. The vision-related components (ImAcqService and ImProcNode) run on the same physical machine (VIS). This is motivated by the substantial cost of image transmission over the network. ImAcqService performs acquisition from C1, C2, C3. C1 is the camera with the 35 mm focal length lens employed for close-range measurement, and it is used to perform the experiment described further.

Vision system for sharpness measurement

In order to realize robot movement with visual feedback on how focused the tool plate is, the following system is employed. A simple planar calibration object is attached to the robot’s tool plate (Fig. 6). The object constitutes a series of rectangles of different sizes enclosed one inside the other. A rectangle’s border is a thin black line. The motivation is that when the object is out of focus, the thin lines become blurred. By devising a method that systematically measures sharpness of the object, it can be possible to provide the necessary feedback to MasterControlNode. The white background makes the object easily segmentable.

The developed algorithm for sharpness measurement is visualized as an EPypes computational graph in Fig. 7. Here, ellipses represent data tokens, and rectangles represent processing functions. The shaded tokens are the pre-defined configuration values.

Figure 6 Calibration object on the robot’s tool plate.

Figure 7 Computational graph of the sharpness measurement algorithm.

The original image (image) is supplied in the grayscale format. First, it undergoes thresholding operation (highlight_light) to highlight the light regions on the image, including the white background of the calibration object. The thresholded binary image (thresholded_light) is eroded to remove the influence of the black lines in well-focused images (erosion). Further, connected components are identified (find_ccomp). The goal is to segment the connected component belonging to the calibration object. To do that, a filter based on width-to-height ratio range and minimal region area is applied (identify_object_region). The selected region of interest is cropped (crop_roi) from the original image, and is used as an input to the Sobel operator, applied in the x direction (sobel_x). From the middle, in terms of y axis, of resulting gradient image sobelx_im, a horizontal line profile profile_sx is extracted (h_profile_sobelx). As a measure of sharpness (measure_sharpness), the standard deviation of this profile is used: the more the original image is in focus, the greater variability between dark and bright pixels in the calibration region.

Figure 8 demonstrates the intermediate results (sobelx_im and profile_sx tokens) of the sharpness measurement routine for images with different positioning of the calibration object. Poses labeled {1, 2, …, 5} are ranged in decreasing distance to the camera. Pose 3 results in the sharpest image, which is reflected in the highest volatility of profile_sx.

Figure 8 Visualization of sharpness measurement intermediate results for images with different positioning of the calibration object.

For another sequence of 20 pose increments, where the robot arm was gradually moved from the approach pose closer to the cameras, the value of the sharpness token is visualized in Fig. 9. The red vertical line correspond to the image with maximal sharpness (as perceived by a person). It is clearly seen that the sharpness curve, measured with the proposed algorithm, reliably corresponds to human perception.

Figure 9 Sharpness measurement on a sequence of images after consecutive robot motions.

Coroutines

The logic of the focus plane calibration is realized with a set of coroutines. The first two can be regarded as “helper” coroutines. The init_move coroutine performs the initial set of motions: retract to the home pose ξhome, and then transition to the approach pose ξappr:

async  def  init_move(rc):      return  await  rc.cmdexec(           rcommands.MoveJoints([0, −90, 180, 0, 90, 0]), # "home" joint               configuration           rcommands.MoveRelJoints([−90, 60, 30, −90, 0, 0]),           rcommands.MoveRelTool([40, −25, 185, 0, 0, 0]),           rcommands.MoveRelJoints([0, 0, 0, 0, 0, 1.5])      )

The parameter rc of the coroutine supplies reference to the instance of RobotClient. As can be seen from the code, init_move consists of only one await call to the rc.cmdexec coroutine method, with the first command corresponding to the motion towards ξhome, and the latter three commands corresponding to the motion ξhome → ξappr.

The second coroutine, move_tool_z, combines sending the setting speed command and the commands for tool motion in the z direction:

async  def  move_tool_z(rc, appr_speed,  delta_z):      return  await  rc.cmdexec_joined(           rcommands.SetSpeed(appr_speed),           rcommands.MoveToolZ(delta_z)      )

Note that for the increment towards the focus plane, two commands are supplied to the cmdexec_joined coroutine method of the instance of RobotClient. The SetSpeed command produces a single message, while MoveToolZ produces two messages (one for motion and one for break). By supplying them to cmdexec_joined, all three messages will be sent to the RobotServer conceptually at the same time.

The primary coroutine, ufloop, realizes the overall logic of approaching the focus plane with vision feedback. In addition to rc, it accepts pbcomm, an instance of ProtobufCommunicator, as well as the parameter for approach vector increment (delta_z, mm) . The source code of ufloop is presented below:

async  def  ufloop(rc, pbcomm, delta_z,  appr_speed):      await  rc.connect()      await  init_move(rc)      sharpness = []      while  True:           pb_req  =  create_vision_request()           await  pbcomm.send(pb_req)           pb_resp  = await  pbcomm.recv()           resp_attrs  =  get_attributes_dict(pb_resp.attributes.entries)           s =  resp_attrs['sharpness']           sharpness.append(s)           # Break  the  loop  when  sharpness  increases           if len(sharpness) >  1 and (sharpness[−1] <  sharpness[−2]):                break           await  move_tool_z(rc, appr_speed,  delta_z)      await  move_tool_z(rc, appr_speed, −delta_z) # retract  back      return  sharpness

First, a TCP connection to RobotServer is established and the initial motion commands are executed. Following that, a series of vision requests are announced using pbcomm. A response obtained as a result of each request contains, amongst others, a real-valued attribute with sharpness measurement from the vision pipeline. A list of these measurements is maintained, and a new value is compared to the previous one. It is expected that sharpness should rise as the robot arm approaches the focus plane. If the new value is smaller than the last, it is an indication that the focus plane has been passed. The robot tool should move one delta_z backwards and the loop should be completed. The final pose of the robot arm can be recorded as ξis.

Coroutines are scheduled to be executed by an event loop. In a simplified form, this includes instantiation of the event loop, the coroutine object (ufloop_coro), and scheduling the latter in the event loop:

loop = asyncio.get_event_loop() ufloop_coro  = ufloop(rc, pscomm,  delta_z) loop.run_until_complete(ufloop_coro)

Time measurements

A MasterControlNode with logic defined with ufloop has been deployed to MCN and run together with an Adept Viper s850 robot and the network configuration described in the “Network topology design.” The vision pipeline from the “Vision system for sharpness measurement” is deployed to the VIS node.

To study the timing properties of such AsyncIO-based logic, the approach phase is of interest (implemented as an infinite loop in ufloop with the sharpness-based termination condition). Figure 10 shows two iterations in the approach phase. Vision (orange) and robot (blue) requests are shown in a way similar to a Gantt chart: a horizontal bar starts at a time instant when the request is sent and ends when the response is received. All time instants are measured in terms of MCN’s AsyncIO clock.

Each iteration i in the approach phase is comprised with a request to the vision system, followed by sending two commands (embodied in the three messages) to the RobotServer. The event-driven nature of the AsyncIO-based implementation is clearly seen for the case of sending the three messages for each robot motion increment. The bytes of the messages are sent asynchronously, and the responses tagged with the same IDs are received as fast as they arrive. RobotServer executes the set_speed and move_rel_tool skills quickly, sending the corresponding responses back to MasterControlNode. Invocation of break blocks until the physical motion has completed.

Figure 10 Vision (orange) and robot (blue) requests during the approach phase of the ufloop coroutine.

For each iteration i, let t(vreq) and t(vresp) correspond to the timestamps of sending the vision request and receiving the vision response respectively (shown as the start and the end of the orange bars in Fig. 10). Similarly, let tssrreq, tmrtrreq, and tbrreq denote timestamps of requests to the RobotServer corresponding to the set_speed, move_rel_tool, and break messages (in Fig. 10, these are shown as starts of the respective blue bars). Notification about the completed motion is received at tbrresp (corresponding to the end of the blue bar for the break command).

Travel time of a robot motion increment τr is measured as a difference between the duration of waiting on the client side and the duration of the operation on the server side. More concretely, it is defined as follows: (1) τr=tbrresp−tssrreq−tbrstart−tssrstart

where tbrstart and tssrstart denote the starting timestamps (measurement in the robot controller’s clock) corresponding to the break and set_speed skills respectively.

Figure 11 shows a histogram of τr measurements derived from an experiment with 32 runs of ufloop program (35 runs, three of which resulted in corrupted data). The appr_speed was varied among the values of {5, 25, 50, 75, 100}. The delta_z parameter was varied among {1.5, 2.0, 2.5}. It can be seen that τr has Gaussian distribution with mean of 5.219 ms and standard deviation of 0.964 ms.

Figure 11 Distribution of travel times of robot motion increments.

The switching durations between robot and vision requests include the time τir→v elapsed between the end of robot requests and start of the vision request, and the time τiv→r elapsed between the end of the vision request and start of the first robot request (set_speed): (2) τir→v=tivreq−tb,i−1rresp

(3) τiv→r=tss,irreq−tivresp.

Here i and i − 1 represent the current and the previous sets of requests, each starting with the vision request, followed by the three robot requests (refer to Fig. 10 and the source code of ufloop for clarification).

Given the same experiment, the histograms representing distributions of τir→v and τiv→r are presented in Fig. 12, with the vertical lines representing the respective minimal and maximal values. The distribution of these characteristics is heavy tailed, with most of the occurrences being around 1 ms.

Figure 12 Distributions of switching durations: (A) between a robot request and a vision request; (B) between a vision request and a robot request.

Durations between the events of starting consecutive send operations for each robot motion increment, i.e., τ(ss→mrt) for set_speed →move_rel_tool and τ(mrt→b) for move_rel_tool → break, are defined as follows: (4) τss→mrt=tmrtrreq−tssrreq

(5) τmrt→b=tbrreq−tmrtrreq.

The distributions of these durations are visualized in Fig. 13. One can notice similarities with distributions of robot/vision switching durations in Fig. 12, although the range of observed measurements tends to be smaller (with most of occurrences being around 0.2 ms).

Figure 13 Distributions of durations between starting consecutive send operations to RobotServer: (A) set_speed→move_rel_tool; (B) move_rel_tool→break.

The similarity of the distributions of various switching durations visualized in Figs. 13 and 12 reveals the temporal properties introduced by the AsyncIO-based design. Such distributions can naturally be modeled with log-normal PDFs, which in turn could be useful for formalized reasoning about timing uncertainty of the developed applications.

Composition of coroutines

The code of the ufloop coroutine looks simple and clear, despite the built-in communication and asynchronous logic. One can argue, however, that the sequential nature of this particular example could be exploited with more traditional blocking I/O primitives. One argument in favor of coroutines is the resulting more efficient runtime, as shown in Fig. 10. Another advantage is the one of composability: several coroutines can be composed together in a well-defined way and treated as a collection of concurrent “lightweight threads”. For example, a particular event of interest can be continuously monitored by the dedicated coroutine, say listener_coro. It can be scheduled together with ufloop_coro as follows:

loop.run_until_complete(      asyncio.gather(           ufloop_coro,           listener_coro      ) )

The same composition logic can be applied to any number of concurrent coroutines. As synchronization mechanisms, one can use the AsyncIO-native Event, Queue, Condition, and others. Concrete use cases of multiple coroutines composition will be investigated in the further work.

Discussion

Similar work

Establishing flexible interfaces to industrial robot controllers is a widely practiced endeavour, specifically in research environments. Such projects are motivated by the constrained capabilities of the proprietary robot platforms, in particular when it comes to sensors integration, multi-robot synchronization, connectivity with external systems, and utilization of methods and tools from modern software engineering in the robotics domain (Angerer, Hoffmann & Schierl, 2013). Depending on the intended application and the available robots platforms’ capabilities, the developed interfaces offer either fine-grained real-time control, or coarse-grained control with soft real-time properties.

Sophistication of the available robot interfaces vary depending on the original robot platform capabilities and the degree of involvement of the respective robot vendors in the development process.

A great deal of flexible robot interfaces development has been done with KUKA robots. Research around KUKA Lightweight Robot (iiwa) resulted in a fast real-time interface based on UDP and ability to access the controller’s motion kernel in a highly fine-grained manner (e.g., supplying custom trajectories and realization of custom cyclic control modes, such as impedance control), having cyclic time frame of 1 to 100 ms (Schreiber, Stemmer & Bischoff, 2010).

Traditional KUKA robots can be supplied with the vendor-provided Kuka.RobotSensorInterface (RSI) and Kuka.Ethernet KRL XML packages, which require additional investments. The user are nevertheless required to create custom communication solutions integrated into existing robot controller code. An example of such approach is an RSI-based configuration for adaptive robot-based fabrication integrating a CAD system and a 3D sensor (Sharif, Agrawal & Sweet, 2017).

An alternative approach for the traditional KUKA robots is based on application of KUKAVARPROXY, a Windows binary that can be deployed to the Windows-side of the robot controller and thus providing access to global variables using the CrossCom interface (Eriksen, 2017). JOpenShowVar (Sanfilippo et al., 2015) is a Java-based client to KUKA robot controllers via an existing KUKAVARPROXY. It uses TCP/IP to communicate with KUKAVARPROXY, and is hence constrained by soft real-time tasks.

ROS interface for COMAU robots providing position and velocity controller is described by Stefano et al. (2014). The available control modalities include additional and absolute position control, additional current control, trajectory management and modification of pre-planned trajectory. The interface, which is implemented as a multithreaded solution, requires an external PC with real-time Linux. The resulting UPD-based real-time communication with the robot controller is characterized by the time frame of 2 ms.

A comparison of three custom interfaces for direct joint control of NACHI, KUKA, and Universal Robots (UR) is done by Lind, Schrimpf & Ulleberg (2010). Communication with NACHI and UR is based on UDP (with cyclic time frames of 10 ms and 8 ms respectively), while with KUKA –on TCP-based RSI (having the time frame of 12 ms).

Comparing with the abovementioned robot interfaces, pyadept/AdeptServer in its current form stands in the category of coarse-grained soft real-time solutions, and bears most similarity with KUKAVARPROXY and JOpenShowVar. In contrast to the latter, the proposed architecture provides a greater decoupling of higher-level logic with native robot controller logic. When it comes to the Adept robot platform, the proposed solution is the first publicly available flexible interface. Earlier attempts were made around the year 2012 with establishing of a ROS interface for Adept robots (Willow Garage, 2012). However, the limited initial functionality has not been further developed since 2013 (ROS Industrial, 2013). Another novelty of the proposed solution is utilization of AsyncIO coroutines for specification of communication-heavy robot logic.

Lessons learned

The presented solutions for flexible coarse-grained control of Adept robots resulted from the ongoing work on integrating an Adept Viper s850 robot with distributed vision systems based on GigE Vision. The Python codebase evolved together with V+ codebase to form pyadept and AdeptServer respectively. The choice of AsyncIO coroutines enabled specification of composable logic with well-defined coordination of multiple connections.

The choice of Python for implementation of the high-level part of the robot interface may seem questionable, as the real-time guarantees cannot be provided. However, for the purpose of coarse-grained control with communication over TCP/IP, such choice is acceptable. The current version of the system is designed to provide reasonable timing characteristics and assure logically correct robot behavior and flexible interaction with external systems. The latter aspect motivated the application of Python coroutines to allow specification of complex tasks based on composition.

During development of the presented solutions, the biggest hurdle was associated with implementation and debugging of V+ logic. As the V+ language is designed for implementing robotic tasks directly in the robot controller, it suits well for use cases with repeatable pre-defined motions. At the same time, V+ debugging capabilities are extremely limited for non-trivial tasks. One of these is communication with external systems. Networking capabilities are built-in in V+, although implementation of specific clients and servers is error-prone.

Big advantage of the proposed architecture is that the robot controller is treated as a service: once the server task is launched, the central application logic is driven by the Python-based MasterControlNode. This comes in hand particularly during interactive development: more complex robot scenarios can be realized in a high-level programming language quickly, with a rich set of communication capabilities.

It is clear that the proposed architecture cannot beat a dedicated V+ program in terms of execution speed. For one thing, the communication overhead exists per each command. Secondly, because the RobotServer maps the incoming ASCII byte strings to the available skills (realized as V+ programs) and executes the latter with the CALLS call (a V+ instruction allowing to call a subroutine identified by its string-based name and a sequence of arguments), the operation is by definition slower than if V+ programs would be called natively. On the other hand, the proposed architecture targets applications with discrete event behavior, where no fine-grained control is required, and the overall sequential logic of the operation and communication capabilities are of a greater importance.

As mentioned before, the most challenging part of development of the presented solutions lies in ensuring that V+ code in AdeptServer works correctly and reliably. As such, the forthcoming research and development efforts shall be focused on improvement of the V+ functionality.

Summary and Further Work

This paper has presented an industrial robot control architecture for the Adept V+ platform that aims at maximizing flexibility of system development. The latter is achieved by explicitly incorporating event-based logic in the master control node and use of AsyncIO as the underlying platform. By creating modular logic blocks as coroutines, one achieves well-defined composability and sophisticated networking capabilities. On the robot controller side, a native V+ server and a collection of robot skills are realized. As such, in the proposed architecture, the robot controller is treated as a service, with all the core logic being implemented on a higher level and with extensible communication capabilities.

In its current form, pyadept, the library incorporating the ideas described in this paper, can be used on any platform supporting the latest versions of Python (3.6 and onwards) and TCP/IP connectivity. When used together with the RobotServer code, pyadept allows to prototype robotic applications that communicate with other distributed components via TCP/IP or ZeroMQ. It can also be extended with additional networking modalities, due to the use of the polymorphic AsyncIO’s reader/writer interface.

The presented solutions were validated on an application combining robot motion with vision feedback. Experiments based on this application revealed timing properties of data transmission and AsyncIO-driven task coordination.

When compared with the established robotic frameworks, in its current form, pyadept is platform-specific and rather high-level. The goal, however, is not to develop yet another robotic middleware, but to test-drive the ideas of using asynchronous coroutines and event-driven logic in building robotic systems in distributed environments. The most natural future strategy is to adapt both the Python code in pyadept and V+ code in AdeptServer to be used with ROS2. The latter natively supports Python 3, so the available codebase serve as a framework for coroutine-based ROS2 node design. The V+ code can form the basis for Adept robots support in ROS2. Before proceeding with ROS2 integration, it is beneficial to quantitatively compare performance of a more complicated robotic application as implemented based on ROS/ROS2 primitives vs pyadept.

In future iterations of the proposed architecture, it is of interest to investigate more high-performance approaches, such as binary message formats and high-frequency periodic UDP-based communication.

List of abbreviations

ACE (Adept ACE) Automation Control Environment

ACE (C++ toolkit) The ADAPTIVE Communication Environment

AVG Automated guided vehicle

AMQP Advanced Message Queuing Protocol

API Application Programming Interface

ARM a processor architecture

ASCII American Standard Code for Information Interchange

BSD Berlekey Software Distribution

CAD Computer-Aided Design

CERN The European Organization for Nuclear Research

FIFO First in, first out

FxIS Flexible Image Service

ID Identifier

IEEE Institute of Electrical and Electronics Engineers

IOCP Input/output completion ports

IP Internet Protocol

IoC Inversion of Control

KRL KUKA Robot Language

LAN Local Area Network

MQTT Message Queuing Telemetry Transport

ODP Oldest packet drop

PDF Probability density function

RSI Robot Sensor Interface

TCP Transmission Control Protocol

UDP User Datagram Protocol

VLAN Virtual Local Area Network

YARP Yet Another Robot Platform

ZMTP ZeroMQ Message Transport Protocol

ROS Robot Operating System

XML eXtensible Markup Language

Additional Information and Declarations

Competing Interests

Author Contributions

Data Availability

The authors declare there are no competing interests.

Oleksandr Semeniuta conceived and designed the experiments, performed the experiments, analyzed the data, contributed reagents/materials/analysis tools, prepared figures and/or tables, performed the computation work, authored or reviewed drafts of the paper, approved the final draft.

Petter Falkman authored or reviewed drafts of the paper, approved the final draft.

The following information was supplied regarding data availability:

The pyadept source code is available at https://github.com/semeniuta/pyadept.

The AdeptServer source code is available at https://github.com/semeniuta/AdeptServer.

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
