# Peer review of "Event-driven industrial robot control architecture for the Adept V+ platform"

_PeerJ Computer Science, doi:10.7717/peerj-cs.207_

## Round 0.1 · original submission · Minor Revisions

Please revise according to the reviewers comments.

Reviewer 1 ·

Basic reporting

The paper presents an event-based system/architecture for the control of Adept robots. The system is written in Python and is based on Python coroutines and the asyncio library.

The introduction and background sections of the paper is nicely written and give a good overview of event-based vs. periodic control, events in computational systems, event-based middleware, and related work on software for building complex robotic systems, e.g., ROS and YARP.

The language is good throughout the paper. However, some minor points need to be addressed: Words like 'the', 'a', and 'an' are missing in many sentences. A native English reader should go through the paper.

Experimental design

An application use case of the system is presented. A simple vision-based inspection system is implemented using OpenCV, where the robot should pick up an object and present it in front of the camera.

The main part of interest in the experimental section is the presented listings in the section “coroutines”. This way to program robots is quite nice.

The section on time measurements show that the system is capable of non/soft real-time control, however, the lack of comparison with a similar implementation in, e.g., ROS, makes it impossible to see if the proposed system performs better or worse than that.

I would have liked to see a comparison of the presented system with an implementation of the system in ROS. With the authors' obvious knowledge of robots and Python programming, I suspect that this should not take them longer than a couple of days. However, this is not a requisite for publication, in my opinion.

Validity of the findings

The presented way to program robots using Python coroutines is as mentioned quite nice. However, I do not see why researches should switch from using ROS to this system as the paper stands now. This could have been clarified with a direct comparison with an implementation in ROS.

Reviewer 2 ·

Basic reporting

This manuscript proposes an architecture that aims to facilitate the interaction between a robotic system, control modules and other external systems. The architecture (methodology) is clearly justified based on the state-of-the-art, the authors provide a strong background supported by literature. A case study is presented as an example of how to implement the architecture in an industrial environment. The language used is appropriate and the content is technically correct.
However, there is a heavy use of acronyms in the manuscript, which is understandable based on its content. I suggest that the authors include a list of abbreviations at the beginning of the manuscript to make it easier for the reader to understand some of the concepts, I suggest some changes in order to help the flow of the paper (SEE GENERAL COMMENTS).

Experimental design

The research conducted follows the Aims and Scope of PeerJ Computer Science. The introduction is strong and clearly describes the research problem that is addressed in this paper. It also describes the knowledge gap and the need for an architecture that solves the problem.
The conducted research is reproducible and the source code is accessible online. Additionally to this manuscript, I encourage the authors to add relevant documentation along with the code. A user’s guide with a case study in tutorial format with examples would be useful for anyone who might be interested in using this Python library. It does not affect this publication but I wanted to take the opportunity to remind the authors to follow good practices when sharing code and the library documentation is missing in this case.

Validity of the findings

The architecture provides a structure that connects multiple elements while easing the communication between them in a robotic environment.
See minor comments on the “system architecture” and “application use case” in general comments. The weakest element is found in the lack of context for the 32 test runs; I highly encourage the authors to document the experiment with a detailed context.
I also suggest trying to restructure the “discussion” and “summary and further work” since I found that some concepts are repetitive and could be merged in one single section: “Conclusions and further work”. An intuitive structure of this section would be: (1) Summary, (2) Architecture highlights, (3) Achieved improvements from existing platforms, and (4) Further work

Additional comments

1. Introduction:
i. There is not need to introduce ROS in the introduction since it will not be used in the architecture.
ii. Make the objectives and contributions clearer. For example, line 95-96 are contributions and should go with the text that finishes in line 74
2. Background:
i. Line 97: reference the software
ii. Lines 101-108: a figure that comes along with this paragraph would be helpful to understand the programming approach
iii. Lines 109-119: again, a figure would be useful. It is hard to follow the text without a diagram that shows how the elements are interconnected
iv. Lines 135-141 and algorithm 1: some variables are not defined. I suggest to at least give a reference in the text or add comments on the algorithm of what each line does
v. Line 148: please add reference to the AsyncIO Python library documentation
vi. Line 198: a reference to the conducted study is missing
vii. Line 202: there is no need to introduce ROS in the introduction since it is described here. I would also remove the reference to ROS2 since is not used in the paper. The mention of ROS2 in future work should be enough
3. System architecture:
i. Line 257: please define what those 6 arguments are. I assume that they are x, y, z and roll, pitch, yaw but they have to be defined with the example similar to lines 271-273.
ii. I would move the examples in page 6 under “Components and general principles” (after line 253) to the “communication protocol” section. It makes more sense since the commands are explained there.
4. Application use case
i. Line 395: typo: it is Figure 4
ii. Lines 412-421: please relate to the corresponding objects and function calls from Figure 6 when explaining how the algorithm works.
iii. Line 422: although Figure 8 might seem relevant to support the algorithm introduced in Figure 6, it is not relevant for the purpose of the paper. The graphs axes are not labeled and I cannot see how it demonstrates the sharpness measurement routine. Please consider to explain the figure in more detail or remove it.
iv. In case that the authors decide to keep Figure 8, it should be placed before Figure 7 in the manuscript.
v. Page 15: the equations must be numbered on the right side of the line. Please indicate what each element in the equations mean; most of them are not formally defined.
vi. Line 525: the conducted experiment needs more context: why 32 runs? how was the experiment conducted? randomly? what are the ranges of appr_speed and delta_z?
vii. Line 528: please indicate what variable is each
viii. Line 532: Please indicate what variable is each
ix. Figures 11 and 12 need more explanation. What does that particular distribution mean to the overall architecture?
x. Line 537: I suggest moving the “composition of coroutines” after the coroutines are defined and before the time measurements study
5. Discussion
i. Line 555: I think it would be make more sense to place “similar work” under the background section as a subsection. The last paragraph can be added to conclusions to highlight the improvements made to existing research

---

## Round 0.2 · accepted · Accept

Please read proof the final version, checking spelling and grammar, possibly with existing automatic tools

Reviewer 1 ·

Basic reporting

no comment

Experimental design

no comment

Validity of the findings

no comment

Reviewer 2 ·

Basic reporting

no comment

Experimental design

no comment

Validity of the findings

no comment

Additional comments

The research concepts and conducted work were valid and strong from the initial version. These minor changes help the reader better understand the contents of the paper. I considered valid the authors' decision of no re-structuring the last part of the manuscript; it's just a writing style preference.
I highly appreciated that the authors followed my recommendation of adding documentation to the Python libraries; it didn't affect the manuscript but I believe it was relevant for the significance of their research.